# Single-cell RNA sequencing reveals midbrain dopamine neuron diversity emerging during mouse brain development

Katarína Tiklová[1], Åsa K. Björklund [2], Laura Lahti[1], Alessandro Fiorenzano[3], Sara Nolbrant[3], Linda Gillberg[1], Nikolaos Volakakis[1], Chika Yokota[4], Markus M. Hilscher[4], Thomas Hauling[4], Fredrik Holmström[1], Eliza Joodmardi[1], Mats Nilsson[4], Malin Parmar [3] & Thomas Perlmann[1,5]

Midbrain dopamine (mDA) neurons constitute a heterogenous group of cells that have been intensely studied, not least because their degeneration causes major symptoms in Parkinson's disease. Understanding the diversity of mDA neurons – previously well characterized anatomically – requires a systematic molecular classification at the genome-wide gene expression level. Here, we use single cell RNA sequencing of isolated mouse neurons expressing the transcription factor *Pitx3*, a marker for mDA neurons. Analyses include cells isolated during development up until adulthood and the results are validated by histological characterization of newly identified markers. This identifies seven neuron subgroups divided in two major branches of developing *Pitx3*-expressing neurons. Five of them express dopaminergic markers, while two express glutamatergic and GABAergic markers, respectively. Analysis also indicate evolutionary conservation of diversity in humans. This comprehensive molecular characterization will provide a valuable resource for elucidating mDA neuron subgroup development and function in the mammalian brain.

[1] Ludwig Institute for Cancer Research, Box 240SE-171 77 Stockholm, Sweden. [2] Department of Cell and Molecular Biology, National Bioinformatics Infrastructure Sweden, Science for Life Laboratory, Uppsala University, Husargatan 3, SE-752 37 Uppsala, Sweden. [3] Developmental and Regenerative Neurobiology, Wallenberg Neuroscience Center, and Lund Stem Cell Centre, Department of Experimental Medical Science, Lund University, SE-221 84 Lund, Sweden. [4] Science for Life Laboratory, Department of Biochemistry and Biophysics, Stockholm University, SE-171 65 Solna, Sweden. [5] Department of Cell and Molecular Biology, Karolinska Institutet, SE-171 77 Stockholm, Sweden. These authors contributed equally: Katarína Tiklová, Åsa K. Björklund, Laura Lahti. Correspondence and requests for materials should be addressed to K.T. (email: katarina.tiklova@ki.se) or to T.P. (email: thomas.perlmann@ki.se)

D opamine (DA) is an essential neurotransmitter in the brain, controlling motor behaviors, cognition, memory, and reward. The major group of neurons releasing DA is situated in the ventral midbrain and projects to more rostral regions in the brain. These neurons referred to as midbrain DA (mDA) neurons, degenerate in patients with Parkinson's disease, and abnormal DA neurotransmission has also been associated with disorders including schizophrenia and addiction. As a consequence, how these neurons develop during embryogenesis, and how they function in the adult brain has been intensely studied. However, since mDA neurons are heterogenous with diverse marker gene expression, innervation targets, and functions, it has not been possible to disentangle this neuron group into clearly defined mDA neuron subtypes. A major challenge is also to identify subclasses of mDA neurons in culture, where information of anatomical location and projection areas are lost.

All mDA neuron subtypes are generated from a common population of proliferating mDA neural progenitor cells localized in the ventral midline of the developing midbrain. After cell cycle exit they initiate expression of neuronal, as well as mDA neuron-specific, genes such as the transcription factor *Pitx3*, migrate to their final destinations within the midbrain, and extend axons toward their projection areas[1,2]. Investigation into mDA neuron development has been a useful model for studies on mechanisms underlying neuron-specific development, but has also been highly motivated by the interest in developing a stem cell replacement therapy for Parkinson's disease (PD) which depends on the generation of mDA neurons in tissue culture from pluripotent stem cells, followed by transplantation to the brain of PD patients[3].

As developing mDA neurons differentiate they diversify into several mDA neuron subtypes with distinct innervation targets and functions[4]. The subdivision into A8-A10 catecholamine nuclei in the midbrain—substantia nigra pars compacta (SNc), ventral tegmental area (VTA) and retrorubral field (RRF)—was initially based on anatomical parameters[4]. However, further analysis of innervation targets, expression of specific genes, and neurophysiology have elucidated additional diversity[5]. Accordingly, both SNc and VTA can be further divided in subdomains with distinct functions and innervation targets. Neurons in the SNc innervate mainly the dorsolateral striatum through the nigrostriatal pathway and control essential motor functions that become severely affected in PD[4]. The neurons of the VTA, which are less vulnerable in PD, project to nucleus accumbens and prefrontal cortex through the mesolimbic pathway and control motivation, cognition, and long-term memory. However, the full extent of mDA neuron diversity remains to be elucidated, and functional properties of distinct mDA neuron subgroups are far from being understood. Furthermore, it still remains unknown when and how this diversity is generated during development, and it is so far not possible to control this subtype diversity during stem cell differentiation.

Recently developed methods focusing on mRNA sequencing (RNAseq) from single cells (scRNAseq) have revolutionized the possibilities for defining distinct cellular states and diversity at the genome-wide gene expression level. Previously, we and others have used scRNAseq to study early mDA neuron development in mouse from E10.5 to E13.5[6,7]. These studies elucidated the transition from proliferating mDA neuron progenitors into early postmitotic mDA neurogenesis. However, at these early developmental time-points mDA neuron differentiation into subgroups has yet not been fully established. Another gene profiling study used qPCR to analyze 96 isolated *Slc6a3* (*dopamine transporter, Dat*)-expressing early postnatal mDA neurons[8]. In addition, a scRNAseq study of developing and adult mouse ventral midbrain cells, also including mDA neurons, analyzed genome-wide gene expression in individual cells[7]. However, both studies sampled a relatively low number of differentiated mDA neurons and in the qPCR study only 96 genes were analyzed.

Here we follow later maturation and continuous generation of gene expression diversity in mDA neurons at high resolution by scRNAseq. We sample postmitotic mDA precursor cells and differentiated neurons from the mouse ventral midbrain from several embryonic stages during mDA neuron maturation, as well as from perinatal and adult mice. This results in a comprehensive transcriptomic map describing continuous mDA neuron maturation into a number of mDA neuron subgroups residing in anatomically defined positions within the adult ventral midbrain.

## Results

**scRNAseq establishes a trajectory for mDA neuron maturation.** Previous studies have shown that the transcription factor *Pitx3* is expressed in all adult mDA neurons[9]. In our studies we took advantage of *Pitx3*[eGFP] mice, a mouse strain harboring the eGFP coding sequence targeted to the *Pitx3* gene locus[10]. We initially analyzed TH and GFP expression pattern in the ventral midbrain of *Pitx3*[eGFP/wt] heterozygous mice (Fig. 1a). Consistent with previous studies[10,11], immunohistochemistry using antibodies against GFP and TH showed that GFP was expressed in virtually all TH-positive mDA neurons throughout the adult mouse ventral midbrain region (Fig. 1a). In addition, cells that were negative for TH but positive for GFP were also identified in the medial VTA. Thus, in addition to mDA neurons, *Pitx3* also appeared to be expressed in cells containing low levels or no TH. An antibody specific to PITX3 was used in immunohistochemistry and confirmed that the PITX3 protein expression closely matched GFP expression in heterozygous *Pitx3*[eGFP/wt] mice, and also confirmed expression in TH-negative cells in the medial VTA (Supplementary Fig. 1a). These cells were also negative for *Slc6a3* expression, as determined by analysis of lineage marked cells using a mouse line expressing Cre under the control of *Slc6a3* regulatory sequences (*Dat*[Cre]; Supplementary Fig. 1b). Thus, the analysis confirmed the localization of PITX3 protein in TH-positive cells, but also identified a population of TH-negative cells positive for PITX3.

Fluorescence activated cell sorting (FACS) was used to isolate GFP-positive cells from dissected ventral midbrain of *Pitx3*[eGFP/wt] embryos and mice from different stages of development up until adulthood (Supplementary Fig. 1c, d). Libraries for scRNAseq were generated using the Smart-seq2 protocol[12]. Following quality control (Supplementary Fig. 2), a total of 1106 cells from embryonic days (E) 13.5, 15.5, 18.5, and postnatal days (P) 1, 7, and 90 were retained in analyses (Supplementary Fig. 1g). A principal component analysis (PCA) considering a gene set of the 710 most variably expressed genes clearly separated cells according to developmental age, with young cells occupying the negative range of principal component 1 (PC1) while the most mature cells (P90) occupied the positive range (Fig. 1b).

We employed *Monocle*[13] to further explore temporal changes in gene expression during postmitotic maturation of mDA neurons. Unsupervised clustering by *Monocle* combined with Samseq[14] identified co-varying genes expressed with distinct temporal profiles over pseudotime across all analyzed cells (Supplementary Fig. 3b, c, Supplementary Data 1). Examples of genes expressed with unique temporal expression profiles at either early, late, or intermediate maturation stages of postmitotic development are shown in Fig. 1c, d. We used fluorescent in situ hybridization to validate temporal expression patterns of mRNAs encoding these three genes (*Cd24a*, *Mkrn3*, *Nrsn2*). As shown in Fig. 1e, *Monocle* correctly predicted the expression of these genes as their temporal expression patterns analyzed by in situ

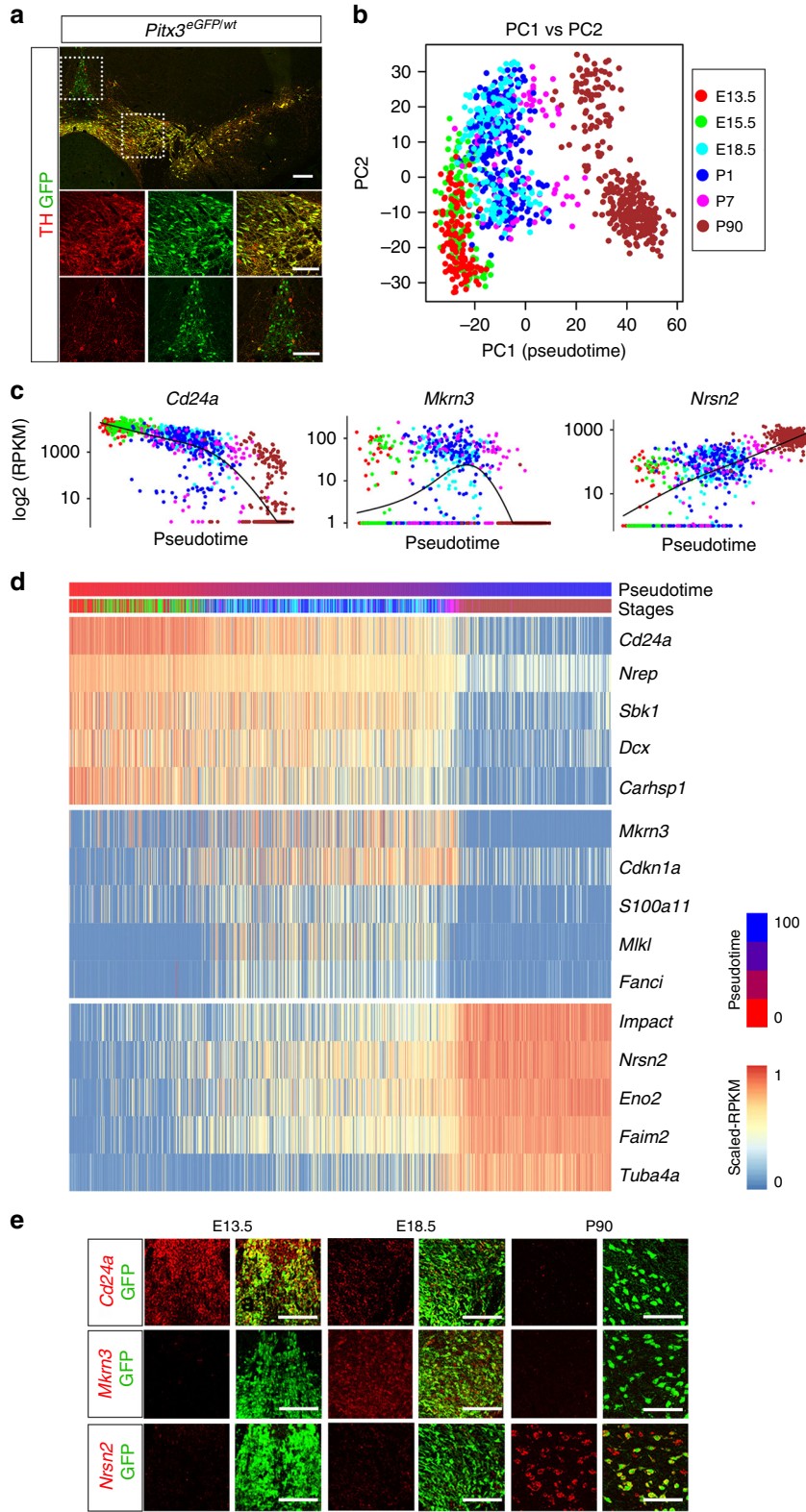

**Fig. 1** Heterogeneity and temporal expression profiles of *Pitx3* cells. **a** Immunostaining analysis of GFP and TH in a frozen section of *Pitx3*^eGFP/wt adult mouse brain. Boxed areas show the localization of the close-ups in the images below. **b** Principal Component (PC) Analysis of the single cells ($n = 1106$). Cells are color-coded by developmental stage. PC1 was defined as the pseudotime axis in remaining analysis with a scale from 0–100. **c** The expression pattern of selected marker genes along the pseudotime. **d** Heatmap visualizing expression of genes related to the maturation of dopaminergic neurons along the pseudotime. **e** Fluorescent ISH of the genes from **c** at different developmental stages. PITX3 expression was detected by GFP staining of the *Pitx3*^eGFP/wt mouse. Scale bars are 100 μm

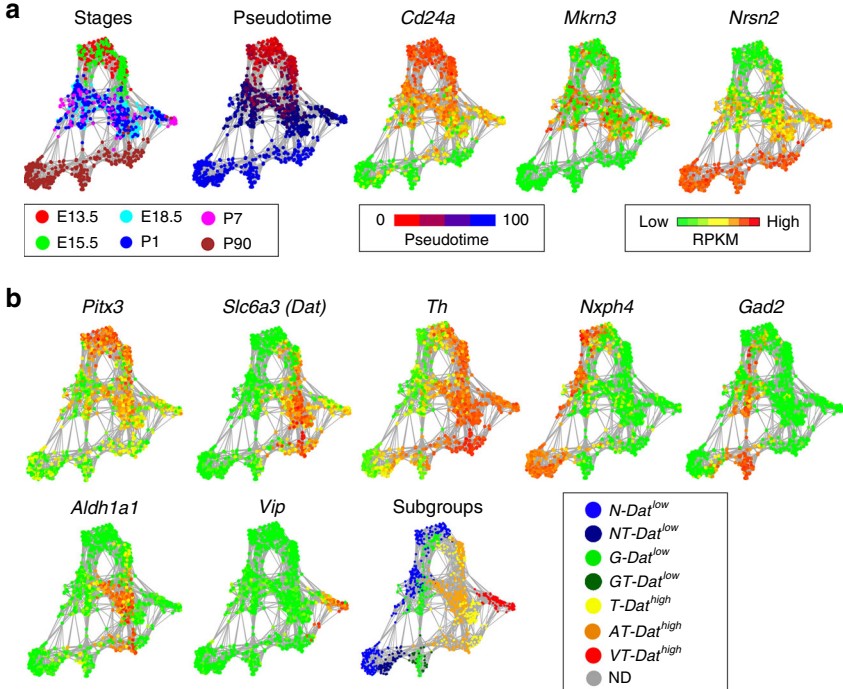

**Fig. 2** Diversity of *Pitx3*-expressing midbrain neurons. **a** Network plot shows a distribution of single cells. The colors indicate the developmental stage of each cell. The expression of *CD24a*, *Mkrn3,* and *Nrsn2* visualized on the network. The colors indicate the RPKM values. **b** The network plot separates the cells into 7 subgroups: N-*Dat*^low in blue, NT-*Dat*^low in dark blue, G-*Dat*^low in green, GT-*Dat*^low in dark green, T-*Dat*^high in yellow, AT-*Dat*^high in orange, VT-*Dat*^high in red and non-defined (ND) cells are in gray. The expression of *Pitx3* and the genes which were used to classify the subgroups (*Slc6a3 (Dat), Th (T), Nxph4 (N), Gad2 (G), Aldh1a1 (A), Vip (V)*) are shown

hybridization peaked at early (*Cd24a*; E13.5), intermediate (*Mkrn3*; E18.5), and late (*Nrsn2*; P90) developmental time points, respectively. *Nrep* and *Sncb* are two additional examples of genes whose temporal expression patterns at early and late stages were validated by in situ hybridization (Supplementary Fig. 3d). Gene ontology terms defined for genes expressed either at early, intermediate or late stages indicated how functional groups of genes are temporally distributed (Supplementary Fig. 3e, f). Thus, the single cell data set provides a resource for mining genes with distinct temporal expression profiles, including genes expressed in postmitotic mDA neurons.

**mDA neuron diversity emerges during postmitotic development**. To identify subclasses of neurons among isolated GFP-positive cells we employed t-distributed neighbor embedding (t-SNE) and graph-based clustering (see Methods, Supplementary Fig. 4a). As illustrated in the resulting cellular network map (Fig. 2a), which organized cells according to transcriptional similarity, a temporal axis was clearly present as illustrated by plotting the expression of early (*Cd24a*), intermediate (*Mkrn3*) and late (*Nrsn2*) gene expression. Distribution of *Pitx3* and *eGFP* were additional examples of genes showing higher expression in early cells and weaker expression in late cells (Supplementary Fig. 4b). Interestingly, two major branches of developing *Pitx3*-expressing cells became evident with low levels of *Dat* to the left side and high levels of *Dat* to the right side of the cellular network (Fig. 2b). These two major branches are referred to as either *Dat*^low or *Dat*^high. Consistent with histological data showing the existence of PITX3-positive cells that were negative for TH, the network plot indicated the existence of maturing cells that expressed either high or low levels *Th*. Cells that expressed low levels of *Th* were mainly included in the *Dat*^low branch of developing *Pitx3*-expressing neurons. Of note, all cells, including the late P90 cells, expressed *eGFP* at the time of sampling as

determined by FACS and reflecting the stability of the GFP protein even when mRNA levels were low (Supplementary Fig. 4b). Although all cells analyzed expressed some levels of *Pitx3* at early developmental stages it is notable that high expression of *Th* clearly correlated with high expression of *Pitx3* (Fig. 2b).

The network map indicated diversity among the analyzed cells and suggested that the analysis may uncover the existence of mDA neuron subtypes. Accordingly, when considering the P90 cells it was evident that the left-hand *Dat*^low cells and the right-hand *Dat*^high cells were subdivided in several subclusters (Fig. 2b). As described, variation along PC1 segregated cells according to temporal maturation (Fig. 1b), but when considering *Nxph4*, *Gad2*, *Dat*, *Aldh1a1*, and *Vip*, genes that contributed most significantly to the variation seen in PC2, PC3, and PC4 (Supplementary Fig. 4c), subgroup trajectories were clearly visualized in the network (Fig. 2b). Thus, *Dat*^low cells could be subdivided into two major developmental trajectories expressing *Nxph4* and *Gad2*, respectively. Moderate or low levels of *Th* expression was seen in roughly half of the P90 *Nxph4* and *Gad2* clusters (Fig. 2b). The *Dat*^low subtypes were thus referred to as N-*Dat*^low (for *Nxph4*), NT-*Dat*^low (for *Nxph4/Th*), G-*Dat*^low (for *Gad2*) and GT-*Dat*^low (for *Gad2/Th*), respectively.

*Dat*^high cells segregated further into two trajectories that are either positive or negative for *Aldh1a1* expression, and one trajectory expressing high levels of *Vip*. The *Dat*^high subtypes were thus referred to as T-*Dat*^high (for *Th*), AT-*Dat*^high (for *Aldh1a1/Th*) and VT-*Dat*^high (for *Vip/Th*). Based on the expression of distinguishing markers, clusters defined by infomap community detection were considered as different subgroups in maturing cells in the network as indicated in Fig. 3a. As noted above, of the seven identified subgroups, five expressed either moderate or high levels of *Th*. Expression of markers enriched for the different subgroups are indicated in Fig. 3b along with a number of typical mDA neuron markers (Supplementary Data 2). Both *Dat*^high and

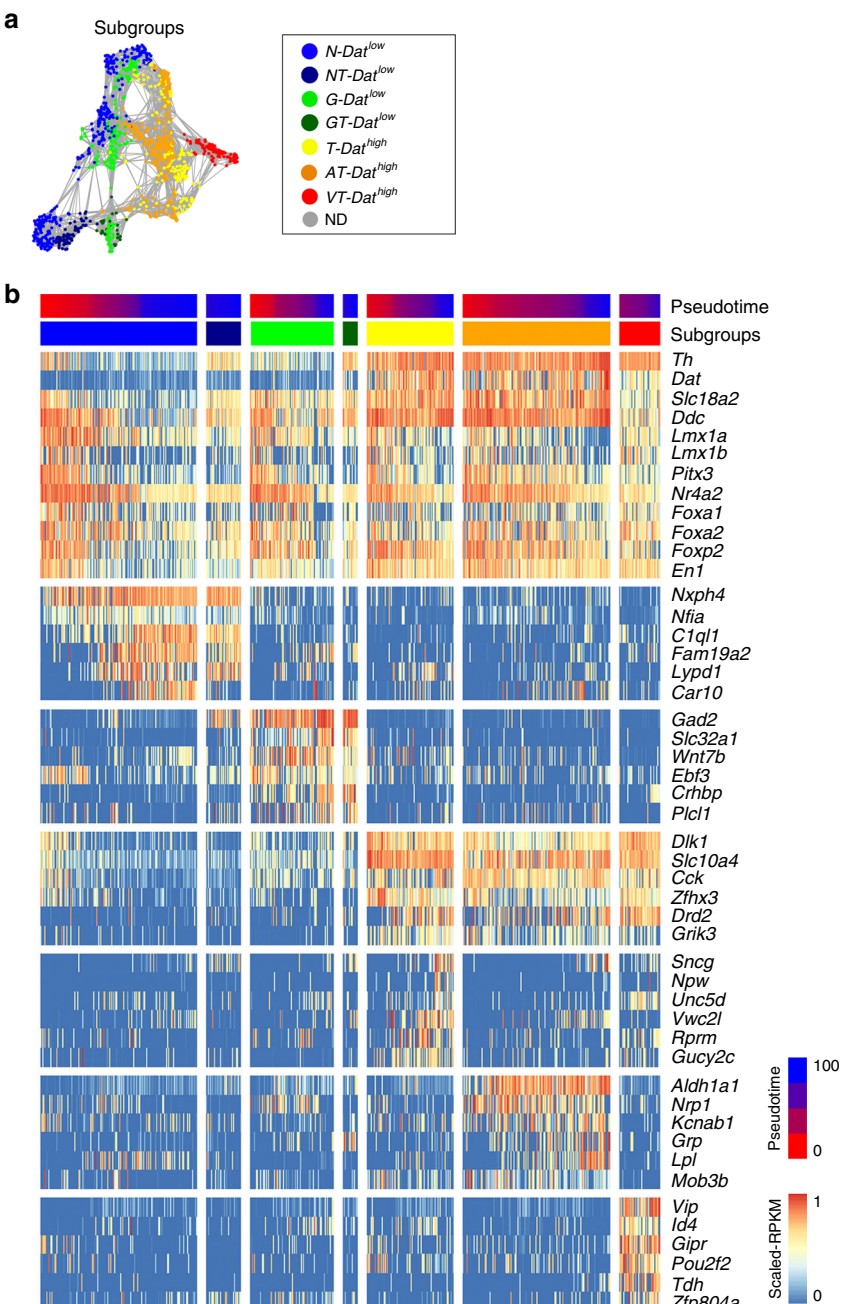

**Fig. 3** Markers enriched in the *Pitx3*-expressing subgroups. **a** The *Pitx3* subgroups visualized on the network (identical to the subgroup schematic represented in Fig. 2b). **b** Heatmap visualizing expression of dopamine-related and subgroup-specific selected genes. The cells of each subgroup are ordered from E13.5 to P90, following the pseudotime

*Dat^{low}* sublineages express common developmental transcription factors including *Lmx1a*, *Lmx1b*, *Foxa2*, and *Nr4a2* (Fig. 3b). Moreover, lineage tracing with *Lmx1a^{CreERT2}* reporter, activated by tamoxifen at E9, demonstrated that both of these cell types originated from *Lmx1a*-expressing neural progenitor cells, thus reflecting a close developmental relationship between all neuronal subgroups described here (Supplementary Fig. 6b). An online application for the analysis of any selected gene in the network is available via the following link (Shiny 2D application: http://perlmannlab.org/resources/).

**Localization of Pitx3-expressing subgroups**. We next determined the localization of the identified *Pitx3*-expressing subtypes within the adult ventral midbrain. Fluorescent in situ

hybridization combined with immunohistochemistry was used to localize distinguishing markers in ventral midbrain tissue sections at different rostro-caudal positions (Fig. 4a). The localization of the different groups is color-coded in the schematic illustrations (Fig. 4g). Immunohistochemistry showed that the *AT-Dat^{high}* cells (co-expressing ALDH1A1 and TH; Fig. 4b) were localized mainly in the SNc, with some cells found also in the paranigral and paraintrafascicular nuclei of VTA[5]. Based on their location and molecular profile, we concluded that *AT-Dat^{high}* cells correspond to mDA neurons innervating the dorso-lateral striatum, which have previously been shown to be vulnerable after exposure to toxins such as 1-methyl-4-phenyl-1,2,3,6-tetra-hydropyridine[15]. In humans the corresponding mDA neurons are the most vulnerable, degenerating at early stages of PD.

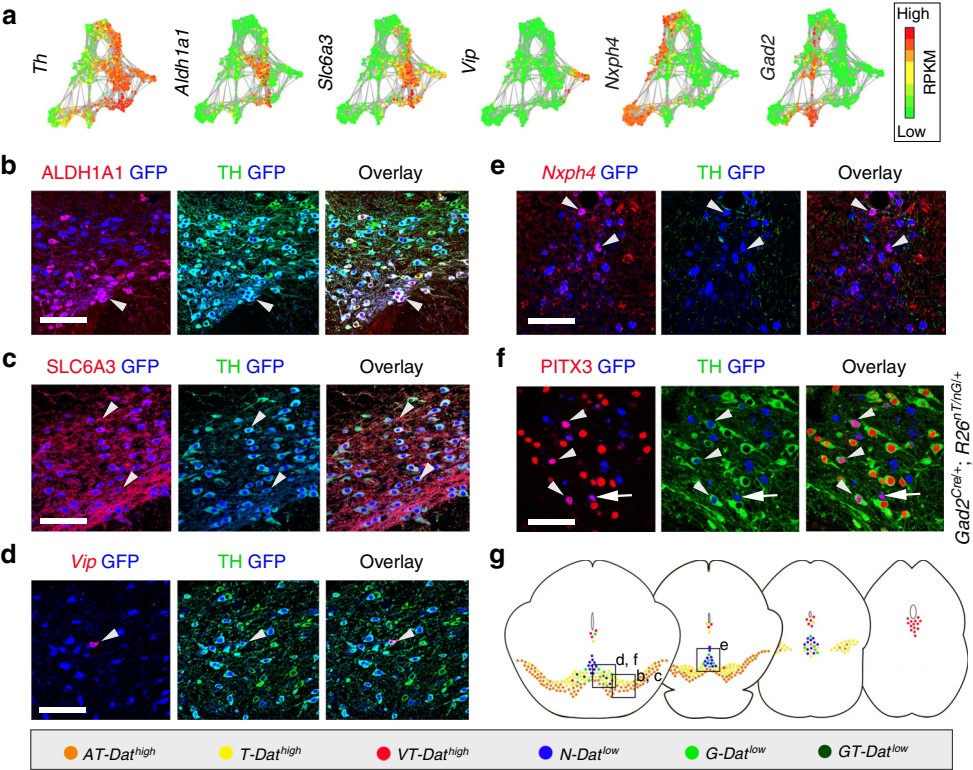

**Fig. 4** Localization of *Pitx3*-expressing subgroups in the ventral midbrain. **a** Network plots of the *Th* and subgroup markers, with green indicating low and red high expression levels. **b–f** Immunostaining (**b**, **c**, **f**) or combined ISH with immunostaining (**d**, **e**) of *Pitx3*-expressing neurons in paraffin sections of P90 *Pitx3^{eGFP/wt}* (**b–e**) or *Gad2^{Cre/+}; R26^{nT-nG/+}* midbrain (**f**). Close-up images are taken from the regions indicated in **g**. Arrowheads indicate examples of co-expressing cells. Arrow (**f**) points to a *Th^{low}* cell and arrowheads to the *Th^{high}* cells. Scale bars are 100 μm. **g** Schematic representation of color-coded *Pitx3* subgroups in the adult mouse midbrain, from rostral to caudal

Additional markers for the *AT-Dat^{high}* cells were validated in Supplementary Fig. 5a. For example, *Sncg* is a prominent marker co-expressed with ALDH1A1 in *AT-Dat^{high}* cells but is also found in *T-Dat^{high}* group (Supplementary Fig. 5b). *T-Dat^{high}* cells (positive for *Dat* and TH, but with no ALDH1A1; Fig. 4c) comprised the main mDA subtype in the parabrachial pigmented nucleus of the VTA but were also detected in the medial VTA. Cells in the lateral parabrachial pigmented nucleus have been shown to project to the lateral shell of nucleus accumbens, whereas medially located neurons innervate nucleus accumbens core and medial shell, basolateral amygdala and prefrontal cortex[16]. Neurons of the *VT-Dat^{high}* group (co-expressing *Vip* and TH) were mainly situated within the periaqueductal gray, a dorsal extension of the A10 group (Supplementary Fig. 5c). This subgroup may correspond to mDA neurons previously implied to participate in the regulation of arousal and wakefulness[17]. In addition, scattered *VT-Dat^{high}* cells were also found within the parabrachial pigmented nucleus (Fig. 4d). In these cells VIP is thought to have a neuroprotective effect against MPTP-induced lesioning[18].

*Dat^{low}* subtypes (*N-Dat^{low}*, *NT-Dat^{low}*, *G-Dat^{low}*, and *GT-Dat^{low}*) were analyzed by the detection of the sub-type distinguishing markers *Nxph4* and *Gad2*, respectively. *Nxph4* mRNA was detected by fluorescent in situ hybridization (Fig. 4e) while *Gad2* expression was detected by GFP immunoreactivity after crossing *Gad2^{Cre/wt}* mice with *R26^{nTnG/wt}* reporter mice (Fig. 4f). Although *Nxph4* expression was relatively widespread within the ventral midbrain, cells co-expressing *Nxph4* and GFP in *Pitx3^{eGFP/wt}* mice were confined to the medially located rostral and caudal linear nuclei of VTA (Fig. 4e). Thus, these locations apparently harbored TH-negative *N-Dat^{low}* subgroup cells. In

these analyses, we were unable to localize *Nxph4*/TH co-expressing cells although scRNAseq indicated the existence of *Nxph4*-expressing cells with moderate expression of *Th* (see Discussion). However, these cells were identified in the in situ sequencing experiment (Supplementary Fig 7b, c). Co-expression of *Nxph4* and *Slc17a6* (*Vglut2*) both by scRNAseq and by in situ hybridization, and the lack of molecular machinery for the generation of DA, indicated that the *N-Dat^{low}* cells could be glutamatergic (Supplementary Fig. 6a). They likely correspond to previously described VTA glutamate neurons projecting to various brain regions, such as nucleus accumbens, ventral pallidum, amygdala, lateral habenula, and medial prefrontal cortex[16]. Indeed, when we injected fluorescent retrogradely labeling beads into the medial prefrontal cortex and nucleus accumbens, we could see few GFP positive, but TH negative, cells labeled in the VTA (Supplementary Figure 8a and see below).

In contrast to the *N-Dat^{low}* subgroup, *G-Dat^{low}*/*GT-Dat^{low}* cells (co-expressing PITX3 and *Gad2* (GFP)) were distributed more broadly in different subnuclei of VTA (Fig. 4f and Supplementary Fig. 6c). These neurons co-expressed *Slc32a1* (*Vgat*), detected both by scRNAseq and in situ hybridization (Fig. 3b; Supplementary Fig. 6c), which indicated that they might use *gamma*-Aminobutyric acid (GABA) as neurotransmitter. Notably, *GT-Dat^{low}* cells (expressing detectable levels of TH) were found only within the parabrachial pigmented nucleus, whereas *G-Dat^{low}* cells (with no detectable TH) were located, in addition to most subnuclei of the VTA, also more dorsally in periaqueductal gray (Supplementary Fig. 6c). The transcription factor encoding gene *Ebf3* is one of the distinguishing markers for the *G-Dat^{low}* subgroup, in particular at late stages (Supplementary Fig. 10a). The *G-Dat^{low}* cells in periaqueductal gray lacked EBF3, unlike

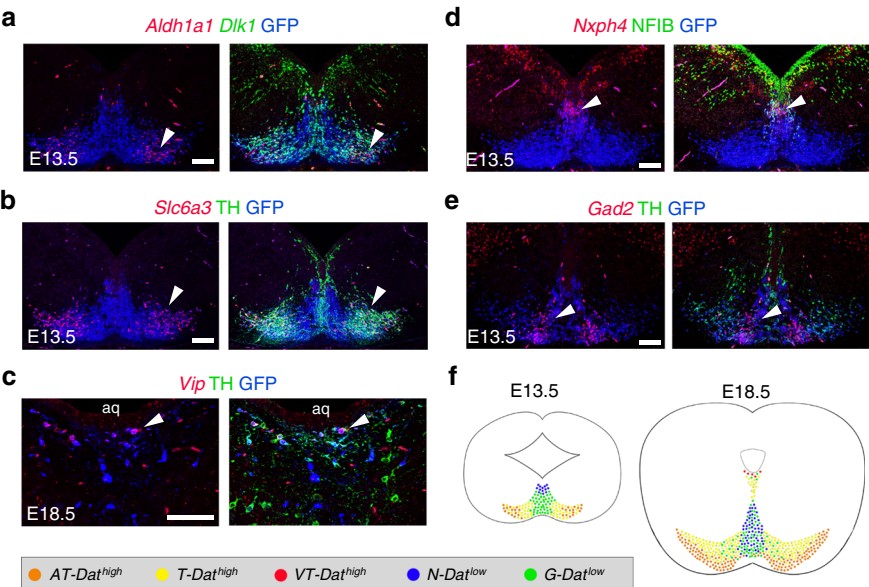

**Fig. 5** The *Pitx3*-expressing subgroups become progressively segregated. **a–e** Close-up views of *Pitx3*-expressing subgroups in the embryonic *Pitx3*^eGFP/wt^ midbrain. Fluorescent ISH with immunostaining on paraffin sections. Arrowheads point to examples of co-expressing cells. Scale bars are 100 μm (**a**, **b**, **d**, **e**) and 50 μm (**c**). Aq, aqueduct. **f** Schematic representation of the color-coded *Pitx3* subgroups in E13.5 and E18.5 midbrain

*G-Dat^low^* cells in other nuclei. The localization of *GT-Dat^low^* cells, and the fact that they co-expressed *Gad2*, *Slc32a1*, and lower levels of *Drd2*, *Dat* and *Slc18a2*, suggests that they might be identical to previously described GABA-co-releasing mDA neurons projecting to lateral habenula[19].

In order to quantify the proportions of cells belonging to each of the seven sublineages, we analyzed *Pitx3*^eGFP^ heterozygous mouse brain sections using an in situ RNA sequencing method[20]. As a result, we were able to map the location of all seven sublineages simultaneously, through the midbrain, at a single-cell resolution (Supplementary Figure 7a–c). Results showed close resemblance to those obtained using traditional ISH and IHC (Fig. 4), further validating the sensitivity of this newer method. The quantification of signals revealed that the majority of cells, 89%, belonged to the three *Dat^high^* lineages, while the four *Dat^low^* subgroups comprised 11% of all Pitx3-eGFP-positive cells (Supplementary Figure 7e). This indicates that the *Dat^high^* groups are significantly underrepresented in the scRNAseq data (see Discussion).

To identify the cell of origin of outgrowth axons within the ventral midbrain, we injected rhodamine-labeled microbeads as a retrograde axonal tracer, into various regions including the main mDA forebrain targets in *Pitx3*^eGFP^ heterozygous mouse brain, with subsequent analysis of labeled cell bodies in the midbrain (Supplementary Figure 8). The results confirmed that the *Dat^high^* cell groups expressing high levels of TH projects to prefrontal cortex, striatum and nucleus accumbens, as previously decribed[21]. In addition, scattered *Dat^low^* cells—identified by low levels of TH —were labeled within the VTA in animals injected with beads to the prefrontal cortex and nucleus accumbens confirming that also these cells project to these VTA targets.

**Progressive segregation of mDA neuron subgroups**. A temporal order of subgroup segregation becomes apparent when considering the network. E13.5 and E15.5 cells cluster closely together, in particular at the earliest time points (Fig. 2b), but cells expressing either high or low levels of *Nxph4* are already segregated to the left and right sides of the plot at early time points (Fig. 2b). Moreover, differential expression of *Dat* and *Aldh1a1* appeared somewhat later during cell maturation since differential expression becomes evident later in development (Fig. 2b). Finally, the *Vip* expressing subgroup trajectory segregates even later.

As expected from the network analysis, the appearance of the *AT-Dat^high^* subtype was evident by immunohistochemistry already at E13.5 and expression of DAT and ALDH1A1 was confined to more lateral mDA neurons (Fig. 5a, b). Also, *Gad2-* and *Nxph4*-expressing subtypes were identified in sections from E13.5 and, as expected, positive cells were found predominantly in the medial developing ventral midbrain (Fig. 5d, e). However, at these early time points distinctions into *G-Dat^low^* versus *GT-Dat^low^* or *N-Dat^low^* versus *NT-Dat^low^* subgroups could not be reliably established. We were also able to detect *Slc32a1* and EBF3 expression—both markers for *G-Dat^low^/GT-Dat^low^* subgroups— in the most medial PITX3-positive cells at E13.5 (Supplementary Fig. 9a, b). At this point EBF3 was still rather broadly detected in the mDA neurons, similar to what could be seen in the network plot in early cells. In addition, and also in accordance with the network plot, *Vip*-expression was not clearly detectable by in situ hybridization until E18.5 in cells of the developing periaqueductal grey (Fig. 5c; see also summary in Fig. 5f).

Thus, the network analysis combined with the histological analysis from embryonic ventral midbrain indicates the existence already at early stages of distinct subgroups that become progressively segregated over time during postmitotic neuron differentiation. To search for candidate regulators that might drive subgroup diversification we explored all annotated transcription factors in the scRNAseq data. Several transcription factors show a distinctive subgroup-specific mRNA expression. As shown in the network (Supplementary Fig. 10a), *Zfp536*, *Nfia*, *Nfib*, *Nfix*, *Tcf4*, *Zfp385b*, *Ebf2*, *Ebf3*, and *Id2* are enriched in *Dat^low^* cells. In contrast, *Zfhx3*, *Pou2f2*, and *Id4* were more highly enriched in *Dat^high^* cells. Further details on mRNA expression in distinctive clusters are summarized in Supplementary Figure 10b. Notably, the majority of these transcription factors are expressed already at early stages indicating that they are candidates for promoting subgroup-specific gene expression.

Previous studies using *Pitx3* null mice demonstrated that *Pitx3* is required for the development of SNc mDA neurons[11]. To investigate this on the molecular level in relation to the seven

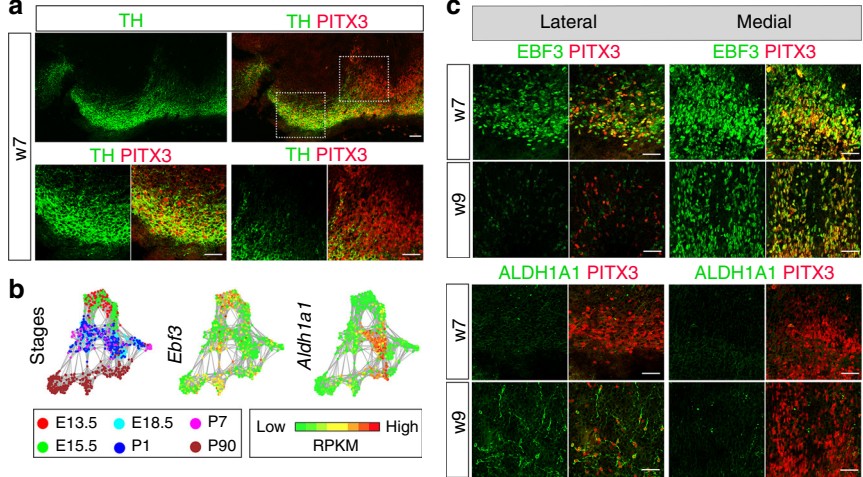

**Fig. 6** The diversity of *Pitx3*-expressing midbrain neurons in human embryos. **a** Immunostaining analysis of PITX3 and TH in human ventral midbrain, on frozen sections. Boxed areas indicate the localization of close-up images. **b** Network plots of the *Ebf3* and *Aldh1a1*, with green indicating low and red high expression levels. **c** Close-up views of medial and lateral regions from ventral midbrain of human embryos which correspond to the boxed regions shown in (**a**). Left sidebar indicates the age of the embryo in weeks. Immunostainings for PITX3, EBF3, and ALDH1A1. Scale bars are 100 μm

*Pitx3*-expressing neuronal subtypes that were identified in this study, *Pitx3* knockout cells from *Pitx3^eGFP/eGFP^* homozygous mice were isolated (Supplementary Fig. 11a) and subjected to scRNAseq. A total of 289 knockout cells from E13.5 embryos and from perinatal (P1) mice were collected and analyzed. The knockout cells were analyzed together with *Pitx3^eGFP/wt^* heterozygous cells from the same stages and by t-SNE network analysis (Supplementary Fig. 11b, c). It was notable that *Pitx3* knockout cells did not form a unique cluster that would indicate an abnormal cellular state. Instead, we found that only few homozygous *Pitx3^eGFP/eGFP^* cells contributed to the *Aldh1a1* and *Dat* subtypes (AT-*Dat^high^* and T-*Dat^high^*) while the other subgroups appeared unaffected by the absence of *Pitx3* (Supplementary Fig. 11d-j). Thus, these results demonstrate how single cell analysis can reveal distinct cell type-specific deficiencies resulting from gene targeting.

**Diversity of mDA neurons is conserved in human embryos.** We next wished to establish if some of the diversity identified in mouse was also seen in human mDA neurons. Immunohistochemistry for visualization of TH and PITX3 expression in human embryonic ventral midbrain at week 7 (w7) revealed a similar heterogeneity as observed in mouse with both high and low TH levels in PITX3-positive neurons (Fig. 6a). The cells expressing high levels of TH were localized laterally, in the SNc and the parabrachial pigmented nucleus, whereas those with low levels of TH were mostly localized more medially, in the rostral and caudal linear nucleus (Fig. 6a). A similar segregation into high and low TH expression in PITX3-positive neurons could also be seen in human embryonic stem cell-derived mDA neurons grafted to the rat striatum (Supplementary Fig. 12).

Since the availability of human tissue material made in situ hybridization impossible to establish for suitable markers, we instead used available antibodies against other markers in immunohistochemistry to investigate additional diversity in human mDA neurons. As seen in Fig. 6b, the scRNAseq analysis showed that the transcription factor *Ebf3* was expressed broadly at early embryonic stages in the mouse (E13.5 and E15.5) but was enriched in *Dat^low^* sublineages at later stages in neurons mostly low in *Th* expression and localized in medial VTA (Supplementary Figs 5c and 6b). The corresponding pattern of expression was found during human embryogenesis with broader EBF3

expression pattern at embryonic week seven, while at week nine EBF3 was only seen in most medial PITX3-positive cells, corresponding to the cells expressing low levels of TH (Fig. 6c). On the other hand, in the mouse *Aldh1a1* was not highly expressed early but was upregulated at later embryonic stages (Fig. 6b). A similar pattern was apparent in human embryos (Fig. 6c). Thus, in seven-week old embryos ALDH1A1 was almost undetectable but at week nine it was expressed in lateral regions, corresponding to TH-high cells. Together these results suggest that a similar diversity develops over time in human and mouse embryonic mDA neurons.

## Discussion

mDA neurons form a diverse group of neurons consisting of several different subgroups with distinct innervation targets and functions. However, classification of mDA neuron subtypes has mostly been based on the anatomical location and projections of mDA neurons within the ventral midbrain, and a more systematic characterization based on molecular properties has remained rather incomplete. scRNAseq now provides a powerful method for defining unique cell types which, in turn, will enable more detailed elucidation of the unique functional properties of various cellular subtypes. Importantly, a better understanding of the molecular landscape defining cell types of therapeutic importance will help to improve in vitro differentiation protocols and quality control of cells intended for cell therapy. Here, we used scRNAseq to molecularly define mDA neuron subgroups from *Pitx3^eGFP/wt^* mice expressing GFP in virtually all mDA neurons, as shown here and previously[10]. In order to establish a molecular framework for the development of mDA neuron diversity at the genome-wide level, we sampled GFP-positive cells from several maturation stages, from early embryonic postmitotic precursors up into fully differentiated adult neurons.

The network analysis defined two major branches of *Pitx3*-expressing neurons: One branch consisted of cells expressing low levels of *Dat* (*Dat^low^*) and a second branch expressed high levels of *Dat* (*Dat^high^*). In total, seven subgroups with distinct profiles of gene expression were defined. Four of these were present in the *Dat^low^* branch: N-*Dat^low^*, NT-*Dat^low^*, G-*Dat^low^*, and GT-*Dat^low^* groups; and three were within the *Dat^high^* branch: T-*Dat^high^*, AT-*Dat^high^*, and VT-*Dat^high^* subgroups. Two of the *Dat^low^* groups lacked detectable levels of *Th* mRNA (N-*Dat^low^* and G-*Dat^low^*), as

well as other molecular components of dopamine-producing machinery, indicating the existence of non-dopaminergic *Pitx3*-expressing ventral midbrain neurons. The developmental relationship between both dopaminergic and non-dopaminergic *Pitx3*-expressing cells was notable as seen by the common expression of several transcription factors typically expressed in developing mDA neurons, and lineage tracing with $Lmx1a^{CreERT2}$ reporter. Thus, the finding of developmentally related non-dopaminergic *Pitx3*-expressing neuronal subtypes reveals entirely novel cellular entities as a result of our scRNAseq analysis.

One of the interesting outcomes of this study was the finding that different subgroups of mDA neurons begin to diversify relatively early, with subgroup-specific gene expression appearing in distinct developmental trajectories already at E13.5. Data mining also revealed a small set of trajectory-specific transcription factors, which are candidate transcriptional determinants for promoting phenotypic differences emerging in differentiating mDA neuron subtypes. Furthermore, we were able to identify clusters of genes corresponding to different mDA neuron maturation stages – information that is useful when assessing mDA differentiation from stem cells in vitro where the developmental stage of the mDA neurons is difficult to assess. Moreover, when including cells from *Pitx3* null mutant mice isolated at different timepoints during development, we found that the generation of *Aldh1a1* and *Dat* subtypes ($AT$-$Dat^{high}$ and $T$-$Dat^{high}$) were highly compromised, while the other subgroups were unaffected by the lack of *Pitx3*.

Although our scRNAseq analysis is a comprehensive analysis of mDA neuron diversity it has limitations and needs to be complimented with other methods for a complete characterization. To localize the cells corresponding to the seven identified *Pitx3*-expressing subgroups, we used a combination of several subgroup-distinguishing markers by classical histological methods, and were able to validate the existence of all except the $NT$-$Dat^{low}$ group, i.e., cells expressing *Nxph4* and *Th*. However, this group was validated using in situ sequencing (Supplementary Figure 7). Thus, it is possible that $NT$-$Dat^{low}$ neurons express *Th* mRNA, but not TH protein. Indeed, such cells in the ventral midbrain have been described[22]. In addition, another technical limitation concerns the relative sizes of different mDA subgroups as determined by scRNAseq clustering as compared to the actual sizes of the corresponding nuclei in the brain. Thus, we noted that the $Dat^{low}$ branch represent more than half of the adult mDA neurons in our clusters, although such cells appear much less numerous in tissue sections, analyzed both with traditional ISH and IHC and by in situ RNA sequencing. This discrepancy is most likely explained by difficulties to quantitatively sample long-projecting TH-expressing SN and VTA neurons, as they are likely to be more sensitive to the mechanical dissociation and FACS. For this reason, using RNA sequencing of individual cell nuclei may be of particular value for further analyses of mature neurons projecting over longer distances. Moreover, any method for scRNAseq library construction has limitations and certain mRNAs might be under- or overrepresented in SmartSeq2 libraries. For example, we were unable to detect high levels of the previously characterized SN-marker *Sox6*[23] in our scRNAseq data set.

In two previous reports analyzing RNA expression in single cells, the authors report the identification of five subtypes of postnatal mDA neurons[7,8]. In both studies, $Dat^{Cre}$ mice were used for lineage marking and cell sorting. As we show (Supplementary Figure 1b), this strategy would sample only $Dat^{high}$, but not $Dat^{low}$, cells analyzed in our study. We suggest that the $AT$-$Dat^{high}$ group corresponds to the groups reported by Poulin et al. to express *Aldh1a1* (DA-1A and DA-2B groups) while the $T$-$Dat^{high}$ group corresponds to the (DA-1B and DA-2A groups). It

is quite evident that the $VT$-$DAT^{high}$ group corresponds to the DA-2D *Vip*-expressing group based on several markers. We correlated similarity between clusters in the recent scRNAseq study[7,8] (Supplementary Figure 13) and found that the five clusters showed a high degree of similarity to the $AT$-$Dat^{high}$ and $T$-$Dat^{high}$ clusters in our study. In addition, unlike the previous unbiased sampling of the entire embryonic ventral midbrain[7], our enrichment-based analysis uncovered subgroup diversity already in the early embryo. We were also able to detect the major branches of $Dat^{high}$ and $Dat^{low}$ mDA types, as well as several subtype markers, appearing in the human embryonic midbrain at similar temporal dynamics and similar in vivo positions as in the mouse.

In conclusion, our strategy to sequence $Pitx3^{eGFP/wt}$ neurons across postmititoc embryonic stages up into adulthood successfully revealed the existence of two main-branches of *Pitx3*-expressing neurons, which in turn are divided into seven subtypes in total—five dopaminergic and two non-dopaminergic. This emphasizes the power of enrichment-based scRNAseq in comparison to unbiased approaches. We expect that our findings, together with the accompanying online-resource (Shiny application: http://perlmannlab.org/resources/) will be a valuable tool for increased understanding of mDA neuron differentiation and subtype differences. Our dataset should provide a powerful resource to improve both in vitro enrichment of therapeutically optimal mDA subgroups using transcription factors that segregate early in the developmental trajectory of the different subtypes of mDA neurons, as well as assessment of overall cell quality and developmental stage of the stem cell-derived mDA neurons. It also provides more specific genetic entry-points for studies of mDA circuitry and function.

## Methods

**Animals**. We used male and female wild type C57BL/6NRj and $Pitx3^{eGFP}$ knock-in reporter mice (Zhao, 2004) on embryonic days E13.5, E15.5, E18.5 and between postnatal days P1-P90. $Gad2^{Cre/wt}$ (Jackson laboratories stock #010802[24],) and $Rosa26^{nT-nG/ nT-nG}$ (Jackson Laboratories stock #023537) were crossed to generate $Gad2^{Cre/wt}$; $Rosa26^{nT-nG/wt}$ mice and embryos. The noon of the day of the plug was considered to be embryonic day (E) 0.5. All experimental procedures followed the guidelines and recommendations of Swedish animal protection legislation and were approved by Stockholm North Animal Ethics board.

**Human tissue**. Human fetal tissue was obtained from legally terminated embryos with approval of the Swedish National Board of Health and Welfare in accordance with existing guidelines including informed consent from women seeking elective abortions. By measuring the crown-to-rump length and neck-to-rump length, the gestational age of embryo was determined.

**Tissue dissociation and FACS sorting**. Ventral midbrain tissue was dissected from $Pitx3^{eGFP}$ reporter mice and dissociated into a single cell suspension as previously described[25] using Neural Tissue Dissociation kit for dissociation of E13.5, E15.5, E18.5 tissue (Miltenyi Biotec); Postnatal Neuron kit for P1 and P7 tissue (Miltenui Biotec) and papain kit for P90 tissue (Worthington). For each developmental stage, several animals have been used. After dissociation, the *eGFP* positive cells were FACS sorted using a BD FACSAria III Cell Sorter.

**Library preparation and sequencing**. Sorted cells were processed using the Smartseq2 protocol[26] to generate the cDNA libraries. Nextera XT DNA library preparation kit (FC-131-1024) using dual indexes (i5 + i7) was used for cDNA tagmentation. The quality of cDNA and tagmented cDNA was checked on a High-Sensitivity DNA chip (Agilent Bioanalyzer). Sequencing was performed on Illumina HiSeq 2000, giving 43 bp reads after de-multiplexing.

**Read alignment and quality control**. Reads were aligned to the mouse genome (mm10) merged with *eGFP* and ERCC spike-in sequences using Star v2.3.0[27] and filtered for uniquely mapping reads. Gene expression was calculated as reads per kilobase gene model and million mappable reads (RPKMs) for each transcript in Ensembl release 69 using rpkmforgenes[28]. The low-quality libraries were filtered out based on following parameters for E13.5, E15.5, E18.5, P1, P7 cells (Supplementary Figure 2a): > 17.1% uniquely mapping reads, < 66% fraction mismatches, > 62% exon mapping reads, < 7.8% 3'mapping, at least 15% of all genes detected,

> 100,000 normalization reads and for P90 cells (Supplementary Figure 2b): > 21% uniquely mapping reads, < 80% fraction mismatches, > 61% exon mapping reads, < 8.8% 3'mapping, at least 3.4% of all genes detected, > 100,000 normalization reads. From 1699 sequenced cells 1562 passed the quality control. After the quality control, another 167 cells were excluded from analysis since they clustered with cells which showed absence of *eGFP* expression and high expression of *Olig1*, suggesting that these cells corresponded to oligodendrocytes. Clustering to remove the non-*eGFP* expressing cells was done for the embryonic and perinatal cells with the same method as described in the section below using t-SNE + igraph (Supplementary Figure 1e). In the adult cells, no clear non-*eGFP* expressing cells were detected (Supplementary Figure 1f). 1395 cells were used for further analysis, of which 1106 were *Pitx3^{eGFP/wt}* and 289 were *Pitx3^{eGFP/eGFP}*.

**Cell population definition.** Biologically variable genes were extracted as those with higher variation than the spike-in RNAs similar as was done previously[29]. Only cells from stages E18.5, P1, and P90 contain spike-ins, so variable genes were extracted from those cells. This gave 328 variable genes for E18.5, 327 variable genes for P1, 825 variable genes for P90 and 453 variable genes when E18.5, P1 & P90 were merged together. The 453 variable genes from all 3 datasets were used for the PCA in Fig. 1b.

Each of the 4 sets of variable genes were used to run t-SNE (Rtsne: T-Distributed Stochastic Neighbor Embedding using a Barnes-Hut Implementation, https://github.com/jkrijthe/Rtsne). Each t-SNE represents slightly different separation of the cells due to the fact that most of the gene variation is seen at different stages. An undirected weighted graph was constructed by extracting the 5 closest neighbors from each of the t-SNEs to create a network with the igraph software (Csardi G, Nepusz T: The igraph software package for complex network research, InterJournal, Complex Systems 1695. 2006. http://igraph.org).

Unsupervised clustering into 48 groups was done with Infomap community detection (Supplementary Fig. 4d)[30]. Each infomap cluster was classified into one of *Pitx3* subgroups based on the expression of the top loading genes from principal components 2, 3 and 4: *Slc6a3 (Dat), Nxph4, Aldh1a1, Vip,* and *Gad2* (Supplementary Fig. 4c) as one of the main subgroup if a majority of the cells in a cluster had high expression (RPKM > 150) of one or more of these genes. Thus, infomap clusters were classified into 5 main lineages. The infomap clusters in the *Nxph4* and *Gad2* lineages were further annotated manually as *Th*-low or *Th*-high giving 7 distinct lineages: N-Dat^{low} (for *Nxph4*), NT-Dat^{low} (for *Nxph4/Th*), G-Dat^{low} (for *Gad2*), GT-Dat^{low} (for *Gad2/Th*), T-Dat^{high} (for *Th*), AT-Dat^{high} (for *Aldh1a1/Th*) and VT-Dat^{high} (for *Vip/Th*). In addition, each cluster was assigned a general age group based on the majority of the cells per cluster as: Embryonic (E13.5, E15.5), Perinatal (E18.5, P1, P7) and Adult (P90).

We used the SAMseq software package to define the maturation and subpopulations-specific genes[14]. Differentially expressed genes for each subpopulation are in Supplementary Data 1 and Supplementary Data 2. Genes were considered significantly differentially expressed if q-value < 0.01.

**Pseudotime analysis.** Principal component 1 in the PCA in Fig. 1b was considered the pseudotime axis, and was scaled from 0–100. Genes that are changing along pseudotime were analysed with the Monocle package[13]. To narrow down the analysis, only genes that are differentially expressed between the age groups: Embryonic, Perinatal and Adult were included (Supplementary Data 1) and were also filtered for protein-coding genes. A curve was fitted for expression along pseudotime using Monocle. The curve matrix was further filtered by removing genes with coefficent of variation squared ($cv^2$) higher than the mean $cv^2$ for all genes across different expression levels which left 2167 genes for further clustering. The genes were grouped with Monocle using k-means clustering into 3 groups. The gene clusters were analysed for biological functions using the Ingenuity Pathway Analysis program (QIAGEN Inc., https://www.qiagenbioinformatics.com/products/ingenuity- pathway-analysis). Only functional categories related to Physiological & System Development functions were selected, and specifically subcategories related to Nervous system development.

**Comparison of data to published data set.** SCEsets for data from ref. [7] were downloaded from https://hemberg-lab.github.io/scRNA.seq.datasets/. The La Manno adult DA cells were extracted and normalized using the Seurat R package[31]. Our adult data and the VT-Dat^{high} cluster were normalized the same way. Differentially expressed genes from both studies (the cell-type-specific expression genes in Supplementary Data 2 of La Manno et al.[7] and differentially expressed genes between lineages Supplementary Data 2) were selected as a gene set. Average expression profiles across that gene set was calculated for our and the La Manno clusters and pairwise Spearman correlations were calculated between them as visualized in Supplementary Figure 13.

**Histology.** For collection of the embryonic tissue, pregnant females were euthanized with $CO_2$ followed by cervical dislocation. The embryos were dissected in ice-cold PBS and immersed in freshly made 4% paraformaldehyde (PFA, Sigma-Aldrich) in PBS for fixation. For the adult brains, the mice were deeply anesthetized before pericardial perfusion using +37 °C PBS, followed by +37 °C 4% PFA in PBS. The adult brain tissue was post-fixed for 3–4 days at room temperature (RT). For

paraffin embedding, the embryos were fixed at RT for 2–3 days, followed by dehydration and immersion in Histosec paraffin (Merck) using automated tissue processor (Leica TP1020). After embedding, 5 µm coronal sections were collected using a microtome (Leica HM360).

For frozen sections, the brains were dissected and post-fixed overnight in 4% PFA at +4 °C, and subsequently cryoprotected for 24–48 h in 30% sucrose after the pericardial perfusion. The brains were serially sectioned on a cryostat (NX70) at 12 µm.

**Immunohistochemistry.** For paraffin sections, the paraffin was removed using xylene and the sections rehydrated, using descending ethanol series, into double distilled water. After permeabilization step in 0.3% Triton-X-100 in PBS for 10 min, the samples were rinsed with distilled water before antigen retrieval by heating in target retrieval solution (DAKO) for 6–8 min in a microwave oven. After the samples had cooled to RT, they were washed twice with PBST (0.1% Triton X-100 in PBS), blocked in PBT (10% donkey or goat serum in PBS with 0.1% Triton X-100), and incubated with primary antibodies overnight at RT. The following day the sections were washed twice with PBST and incubated with secondary antibodies RT for 2–4 h, before washing in PBS and mounting in Mowiol-based mounting medium containing DABCO (1.4-diazabicyclo[2.2.2]octane, Sigma-Aldrich). The nuclei were visualized using DAPI (4′-6′-diamidino-2-phenylindole; Sigma-Aldrich). The list of antibodies is available in the Supplementary Table 1.

For frozen sections, the sections were preincubated for 2 h in blocking solution containing 3% normal donkey sera and 0.2% Triton X-100 in PBS. Primary antibodies diluted in blocking solution were applied overnight at 4 °C. After washes with 0.1% Triton X-100 in PBS, secondary antibodies diluted in blocking solution were applied for 2 h at room temperature. The stained sections were washed with PBS and mounted in Vectashield mounting media. The list of antibodies is available in the Supplementary Table 1.

**Immunohistochemistry of rat and human fetal tissue.** The human fetal tissue was fixed in 4% PFA, embedded in OCT mounting medium and cut at 20 µm using a cryostat. For immunohistochemistry of rat tissue, the brains were fixed in 4% PFA and cut at 35 µm using a microtome.

The human tissue was stained directly on gelatin-coated slides and the rat tissue was stained as free-floating sections, using the same staining protocol. All washing steps were done in 0.1 M phosphate buffered saline with potassium (KPBS) and all incubations were done in 5% donkey serum +0.25% Triton-X-100 in KPBS. For antigen retrieval, the sections were incubated in Tris-EDTA buffer pH 9.0 at 80 °C for 30 min. The sections were blocked for 1 h at room temperature in the incubation solution before applying the primary antibodies overnight at room temperature. Primary antibodies included: HuNu (1:200, EMD Millipore, MAB1281), TH (1:1000, EMD Millipore, AB152), EBF3 (1:500, Abnova, H00253738-M05), ALDH1A1 (1:200, Abcam, Ab24343) and PITX3 (1:500, in-house made). The following day the sections were washed and blocked for an additional 45 min in the incubation solution before the secondary antibodies were applied (all secondary antibodies were purchased from Jackson ImmunoResearch Laboratories and used at 1:200). After incubation with secondary antibodies for 1 h, the sections were washed, mounted and cover slipped using PVA-DABCO containing DAPI (1:1000).

**Fluorescent mRNA in situ hybridization.** The DIG (Roche) or DNP (Perkin-Elmer) -labeled cRNA probes were transcribed from linearized plasmids using manufacturers' instructions. The protocol for mRNA in situ hybridization (ISH) for paraffin sections was modified from[32]. For double ISH, both probes were hybridized simultaneously, but visualized sequentially using anti-DIG-POD (1:800, Roche) and anti-DNP-POD (1:1000, Perkin-Elmer) and TSA Plus Cy3 and FITC kits (Perkin-Elmer). After developing the first signal, the first peroxidase was quenched in 0.3% $H_2O_2$ for 30 min, the samples washed twice in TBST, and incubated in the second peroxidase conjugate, followed by visualization with the second color.

For the combined immunohistochemistry and ISH, the samples were washed in PBST after the visualization of the ISH signals, and then blocked with PBT, followed by primary and secondary antibodies at RT as described above. The list of cRNA probes used is available as Supplementary Table 2.

**Retrograde labeling of neurons.** Red fluorescent beads (Lumafluor Inc.) were injected into the brains of deeply anesthetized *Pitx3^{eGFP/wt}* mice using following stereotactic coordinates: dorsolateral striatum AP +1.0, ML −2.7, DV −2.4; nucleus accumbens AP +1.2, ML −1.1, DV −4.9; prefrontal cortex AP +1.7, ML −0.2, DV −2.5; anterior hippocampus AP −1.8; ML −0.75, DV −1.65; posterior hippocampus AP −3.4, ML −2.3, DV −2.15. The volume of the beads, diluted 1:2, was 0.5 µl. The mice were intracardially perfused with 4% PFA one week after the stereotactic surgery, the brains balanced in 20% and 30% sucrose in PBS, frozen, and free-floating sections were cut at 35–40 µm. The sections were washed with PBS +0.1% Triton-X 100, blocked with 10% heat-treated fetal calf serum, then stained with primary antibodies in RT overnight. After washing with PBS, the secondary antibodies were applied in 1% BSA in PBS and incubated in RT for 2 h. The list of antibodies is available in Supplementary Table 1.

**Image capture and processing**. mRNA ISH and IHC samples were imaged using either Zeiss confocal microscopes LSM5 Exciter or LSM700, and composite images compiled, and the brightness and contrast adjusted, using Adobe Photoshop CC 2017.

**In situ RNA sequencing using padlock probes**. The in situ RNA sequencing experiments were performed as described[20]. In brief, fresh-frozen 10 μm thick, coronal mouse brain sections (Bregma: −2.92 mm, −3.08 mm, −3.28 mm, −3.88 mm) were obtained on a cyrostat and stored at −80 °C until fixation. After pre-fixation 3% (w/v) paraformaldehyde (Sigma) in DEPC-treated PBS (DEPC-PBS) for 5 min at RT, tissues were washed in DEPC-PBS and reverse transcription was performed at 37 °C for 14 hrs. Tissues were then treated with RNaseH at 37 °C for 30 min to make single-stranded cDNAs, followed by padlock probe (IDT Corelville, Iowa; 4nmole, standard desalting, 5 prime phosphorylated; Supplementary Data 3) hybridization and ligation. Rolling circle amplification was performed at 30 °C for 14 hrs. Rolling circle products were detected by using AF750-labeled detection oligo (/5AF750/UGCGUCUAUUUAGUGGAGCC, IDT Corelville, Iowa).

Images were acquired using a Zeiss Axio Imager Z2 epifluorescence microscope (Zeiss Oberkochen, Germany), equipped with a 20x objective. A series of images (10% overlap between two neighboring images) at different focal depths (0.49 μm X 9 Z-stacks) was obtained and the stacks of images were merged to a single image thereafter using the maximum-intensity projection (MIP) in the Zeiss ZEN software. The resulting images were then automatically stitched together into a single image containing the entire scanned area. Finally, the stitched image was used for further image analysis and aligned with the stitched images of each sequencing round.

We used barcoded probes in the in situ RNA sequencing experiments and each gene was given a unique label (Supplementary Data 3). The unique label represents a four-base barcode (hamming distance of at least two between barcodes) consisting of bases A, C, G or T. We executed four rounds of sequencing to decode 49 genes by hybridizing fluorescently labeled probes (A=Cy5, C=Texas Red, G=Cy3, T=AF488). The fluorescence intensity from each of the signals was extracted and all intensity information and coordinates were saved to a.csv file using a custom-made Cellprofiler 2.2.1 Pipeline. Signal decoding (correlating the fluorescent intensity to a gene) was done using Matlab 2018a. Thereby, for each signal and hybridization step the base with the highest fluorescence intensity was extracted and a quality score was calculated (defined as the maximum signal, divided by the sum of all signals). After thresholding, the frequency of each sequence was extracted and based on the 2D coordinates a map of genes was built as well as the signals were assigned to cells. Thereby, cell nuclei were segmented using watershed segmentation on the DAPI channel and expanded.

Regions of interest were drawn onto the tissue section and cells within that region were extracted using the sp R package (https://cran.r-project.org/doc/Rnews/). The cells/genes were further filtered using these criteria: (1) keep only cells expressing *Th, Pitx3* or *eGFP*, (2) remove any gene expressed in less than 10 cells, (3) remove genes *Pitx3, eGFP* and *Nxph4* (*Pitx3* and *eGFP* since they should not contribute to the clustering, *Nxph4* since we saw clear overrepresentation of that gene in a majority of the cells), (4) remove any cell that does not express 3 or more genes (Supplementary Data 4). For classification of the in situ data, we used expression of the same set of genes in the RNA-seq data from adult and the *VT-Dat^high* cluster. Both RNA seq data and in situ data were converted to rank-based matrices.

A random forest was trained using the RNAseq gene ranks and the cluster classification as the response vector using the randomForest R package. and parameters mtry = 15 and ntree = 1000. The cluster prediction for each in situ cell was filtered to only keep predictions with >45% of the votes. All others were set to "none". As results varied for each training of the random forest, 10 different random forests were trained and the consensus predictions of the 10 were used only when more than half of the forests agreed on their prediction.

**Code availability**. Custom-code for extracting fluorescence intensity from Padlock probes is available at https://github.com/Moldia/in_situ_seq.

**Reporting summary**. Further information on experimental design is available in the Nature Research Reporting Summary linked to this article.

## Data availability

Data supporting the findings in this study are within this manuscript, on the website http://perlmannlab.org, or available from the corresponding authors upon request. The RNA-seq data have been submitted to the GEO database under the accession code GSE116138.

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

## Acknowledgements

The authors thank Javier Avila and Belinda Pannagel for expert technical assistance at the CMB FACS facility; professor Juha Partanen and Francesca Morello at the University of Helsinki, Finland, for providing Gad2-Cre-reporter samples and probes for in situ hybridization. We acknowledge the ¨In situ sequencing¨ unit of the spatial-omics infrastructure at the Science for Life Laboratory, Sweden for technical assistance with the in situ sequencing experiments. This project was supported by Knut och Alice Wallenberg Foundation (T.P.), Swedish Research Council (VR; grant agreement 2016-02506, T.P.; 2016-00873, M.P.; 2016-03645, M.N.), Torsten Söderberg Foundation (TP), Svenska Sällskapet for Medicinsk Forskning (K.T.), Sigrid Juselius Fellowship (L.L.), the New York Stem Cell Foundation (M.P.), Hjärnfonden (M.P.), Parkinsonfonden (M.P.). M.P is a New York Stem Cell Foundation Robertson Investigator.

## Author contributions

K.T. and T.P. designed the study. K.T., L.L., A.F, S.N., L.G., N.V., F.H. and E.J. performed the experiments. Å.K.B. performed the bioinformatics analyses. S.N. and M.P. provided the human data. C.Y., M.M.H., T.H., and M.N. performed the in-situ sequencing experiment. T.P., K.T., and L.L. wrote the manuscript with input from all authors.

## Additional information

**Competing interests:** M.N. and T.H. hold shares in Cartana AB, a company commercializing in situ sequencing reagents. The remaining authors declare no competing interest.

**Journal Peer Review Information:** *Nature Communications* thanks Trygve Bakken and other anonymous reviewer(s) for their contribution to the peer review of this work. Peer reviewer reports are available.

