## [Peer Review File · Nature Communications]

Reviewers' comments:

Reviewer #1 (Remarks to the Author):

In this study, the authors performed single cell mRNA sequencing analysis of ventral midbrain cells at different embryonic and postnatal ages. They facs-sorted GFP cells from Pitx3-GFP animals. Pitx3 is a ventral midbrain transcription factor mainly expressed by dopamine neurons. They identified 7 subgroups and 5 of them were dopaminergic. They also performed analysis of human fetal tissues and confirmed a certain degree of conservation of cell diversity in humans.

The techniques used in this paper and the analysis are very well performed. However, the message from this study is rather descriptive and could be clearer. Previous studies using single cell analysis also identified dopamine neuron subgroups but were not adequately compared. The author simply mentioned that it is difficult to compare the data to these studies due to different methodologies. The paper from Poulin et al (2014) identified 6 dopaminergic subgroups and among them, they identified Aldh1a subgroup, and VIP subgroup (similar to this study). In addition, the authors argue that previous study using dat-cre mice in combination with a reporter line is not adequate since they are sampling only Dat-high cells. This argument is not valid since even low level of cre is sufficient to induce recombination of the reporter gene, which expression is not dependent on the level of Cre. Dat-cre mice have been used in multiple studies with very effective recombination (virtually all TH+ neurons in the midbrain). I think the authors need to discuss more in detail these previous reports and either confirm or challenge the identification of dopamine neuron subgroups.

Another important aspect is that they do not provide any evidence that these DA neuron subtypes are functionally different. The authors indicate that particular subgroups (Aldh1a) may correspond to cells projecting to the dorso-lateral striatum or another subgroup that may correspond to cells having multiple projections but they do not show any tracing experiments. It would improve significantly the study if cell markers would be combined with tracing experiments using retrograde tracers or genetic axonal labelling (intersectional).

Minor points:

In the result section, the authors indicate that AT-Dat-high cells were located mainly in the dorsal tier of SNc but based on Fig 4, supplemental fig 4b and report from Poulin et al 2014, AT-Dat high population does not seem to be located in the dorsal SNc but rather in the ventral SNc.

What is the difference between fig 2b last panel and fig3a?

Fig 6c, it is indicated TH low and TH high but they do not show TH staining.

In the discussion they show results about transcription factors but it should be moved in the result section.

Reviewer #2 (Remarks to the Author):

The manuscript by Tiklovà et al. reports an excellent study aimed at identifying and characterizing Pitx3-expressing neuronal subgroups in ventral midbrain.

In the first part of this study, the authors applied scRNAseq to 1106 Pitx3eGFP cells isolated from embryonic and post-natal brains. The authors first analyzed a reliable set of variably expressed genes

to cluster these cells on the basis of developmental stage and then analyzed the expression of genes with distinct temporal expression profiles. Both analyses successfully confirm temporal segregation and stage-specific expression. Then the authors analyzed the scRNAseq data to generate a cellular network map where they integrated developmental stages with transcriptional profiles. They generated a map initially distinguishing sorted cells in those with high or low levels of Dat and then, by integrating the expression of additional genes, the authors identified 7 neuronal subgroups.

Immunohistochemistry analysis in adult brain with markers expressed in the identified subtypes allowed their physical map within the Pitx3-expressing domain. Importantly, based on their topology the authors correlated these neuronal subtypes to specific functions. Then, the authors showed that the subtypes they have identified in the adult brain emerged during embryonic development sometimes at different stages. In this context they also reported the subtype-specific expression of several transcription factors which might induce cell-type specific molecular identity during embryonic development. The next step was to investigate the phenotype of Pitx3KO mice showing the precise cell abnormality of this mutant. In the last section of this study the authors reported that neuronal subgroups with a similar identity could be identified also in the human brain.

In my opinion, this manuscript reports an excellent study. This evaluation is based on the logic approach, methodology employed and general relevance of findings. The manuscript includes an exhaustive amount of data aimed together at the identification of Pitx3-expressing (prevalently dopaminergic) neuronal-subtypes, at their ontogenesis and, further, at their correlation with specific functions.

The establishment of this detailed neuronal map should open new avenues which should provide relevant advance to understand development, identity and functioning of specific dopaminergic cell sub-types. This may be important for human diseases related to altered functioning of mDA neurons. In sum I fully support the publication of this manuscript in Nature Communications.

Reviewer #3 (Remarks to the Author):

In their manuscript titled "Single-cell RNA Sequencing Reveals Midbrain Dopamine Neuron Diversity Emerging During Mouse Brain Development", Tiklova et al. use single cell RNA-seq to identify midbrain dopaminergic (mDA) cell types during mouse development. This study provides genome-wide gene expression information for an important class of neurons in the mammalian brain along with a public web viewer to mine that data. This study builds on a single cell RNA-seq survey of this same brain structure described by La Manno et al. 2016 in Cell by profiling more mDA cells and including an adult mouse (P90). The authors report a set of cell type markers that persist through development, which should aid targeting these cells. However, the authors missed an opportunity to better relate their results to this previously published work as commented on below. Also, more work needs to be done to show that these same types have conserved markers in human and can be found at equivalent developmental stages.

Major comments:

- You argue that there is a conserved molecular program between mouse and human mDA neuron development but you are comparing cells from earlier in brain development in human. Week 7 is approximately E12.5 in mouse and Week 9 is E14 (see La Manno et al. 2016 Cell, Figure 2B). This is particularly problematic for the expression of Aldh1a1 which turns on in human at week 9 (E14) while in mouse is still off at this stage. You could profile human fetal tissue at matched developmental stages to mouse or you could test for spatial segregation of additional subgroup markers that are

persistent throughout mouse development (and hence may be turned on as early as 7-9 weeks in human).

- In the discussion, you state that it was difficult to compare your clusters to two existing datasets from Poulin et al. and La Manno et al. This is a missed opportunity that would further validate the diversity of cell types reported in those studies and potentially reveal more diversity. There are a few techniques available to compare samples from disparate datasets – for example, CCA in the Seurat package from the Satija lab (https://satijalab.org/seurat/immune_alignment.html). You could also take a simpler strategy of correlating the expression of cluster marker genes as shown in La Manno et al. (Figure 2A).

Minor comments:

- Mention that you are studying mouse in the abstract
- You should report RNA-seq QC metrics: distributions of mapped reads and gene detection
- Please comment on cell type selection biases that may have been introduced by single cell dissociations. What proportion of Pitx3+ cells do you label in situ using markers for your 7 subgroups?
- (Figure 1b) You show limited separation of cells at E13.5 and E15.5, and in the Methods you state that variable genes were only selected from 3 ages (E18.5, P1, and P90) to generate the PCA. You should state this in the results that this could explain why you don't see more differences between these two ages. You could also use a method that does not rely on spike-ins to select variable genes from all ages for PCA.
- Please explain in more detail how you selected the marker genes to combine clusters from Infomap. Based on the graphs you plot and the Infomap results, it looks like cells from different ages should form their own clusters. Do each of your 7 subgroups contain a similar number of Infomap clusters. If you run Infomap with more than 5 nearest neighbors (which is quite low), do you get fewer clusters that better match the final 7 subgroups?
- (160-161) If eGFP mRNA levels did not reflect protein levels, then why did you exclude cells on the basis of low eGFP expression before analysis? You showed that some GFP+ cells were PITX3- in situ but you have not shown that these cells correspond to the cells that you filtered. You could find a marker gene of the excluded cells and co-label with this marker in situ.
- (235-236) You point out different distributions of G-Dat_low and GT-Dat_low subclusters in the results, but in Figure 4g you group them together. Can you show in the figure the different distributions or else clarify in the legend text that this is not possible?
- (285-286) In SI Figure 8j you show a massive reduction in AT-Dat_high cells by in situ staining but in SI Figure 8d, this reduction looks more modest. How do you explain this difference between RNA-seq profiling and in situ labeling?

Reviewers' comments:

Reviewer #1 (Remarks to the Author):

In this study, the authors performed single cell mRNA sequencing analysis of ventral midbrain cells at different embryonic and postnatal ages. They facs-sorted GFP cells from Pitx3-GFP animals. Pitx3 is a ventral midbrain transcription factor mainly expressed by dopamine neurons. They identified 7 subgroups and 5 of them were dopaminergic. They also performed analysis of human fetal tissues and confirmed a certain degree of conservation of cell diversity in humans.

The techniques used in this paper and the analysis are very well performed. However, the message from this study is rather descriptive and could be clearer. Previous studies using single cell analysis also identified dopamine neuron subgroups but were not adequately compared. The author simply mentioned that it is difficult to compare the data to these studies due to different methodologies. The paper from Poulin et al (2014) identified 6 dopaminergic subgroups and among them, they identified Aldh1a subgroup, and VIP subgroup (similar to this study).

In addition, the authors argue that previous study using dat-cre mice in combination with a reporter line is not adequate since they are sampling only Dat-high cells. This argument is not valid since even low level of cre is sufficient to induce recombination of the reporter gene, which expression is not dependent on the level of Cre. Dat-cre mice have been used in multiple studies with very effective recombination (virtually all TH+ neurons in the midbrain). I think the authors need to discuss more in detail these previous reports and either confirm or challenge the identification of dopamine neuron subgroups.

We thank the reviewer for valuable comments.

As suggested by the reviewer we have now looked further into the previously published studies and discuss them in the revised manuscript. Poulin et al. used Dat-Cre mice to fluorescent labelling of cells for single cell qPCR analysis. We assumed that the Dat^{low} cells that were found in our study would not be sampled by the strategy used by Poulin et al. However, as the reviewer correctly points out, low levels of Dat-Cre might be sufficient for recombination. We have now addressed this directly by using Dat-Cre mice to lineage mark Dat-expressing neurons. We found that Pitx3-expressing (eGFP-positive) cells in the medial VTA, corresponding to Dat^{low} cells in our study (some of them expressing low levels of TH while others are negative), are not marked and thus would not be included by the approach described in the Poulin paper. Thus, from our analysis of Dat-Cre lineage marked cells, now included in a revised Supp Figure 1b and mentioned on page 7, we suggest that the five groups defined by Poulin et al. correspond to the Dat^{high} cells in our study. It is quite evident that the VT-Dat^{high} (Vip-expressing) group of our study is similar to the DA-2D (Vip-expressing) group in the Poulin et al. study. In both studies the majority of the Vip-positive cells are localized in the periaqueductal grey, although we identified Vip-expressing cells also in the VTA, as we show in Figure 4d. The four remaining groups in the Poulin study are distinguished on the basis of Aldh1a1 expression. Accordingly, the AT-Dat^{high} group appears to be analogous to the Aldh1a1-expressing DA-1A and DA-2B groups, and the T-Dat^{high} group is analogous to the DA-1B and DA-2A groups. However, only a few individual markers are distinguishing the different groups in the Poulin study and it is difficult to determine exactly how similar the different groups are. We have now added text in the Discussion to more clearly point out the relationship between our study and that of Poulin et al. (page 19).

In response to both Reviewers 1 and 3 we have also considered the data in La Manno et al., who used single cell RNA-seq of cells from the mouse ventral midbrain (see our response to Reviewer 3 below).

Another important aspect is that they do not provide any evidence that these DA neuron subtypes are functionally different. The authors indicate that particular subgroups (Aldh1a) may correspond to cells projecting to the dorso-lateral striatum or another subgroup that may correspond to cells having multiple projections but they do not show any tracing experiments. It would improve significantly the study if cell markers would be combined with tracing experiments using retrograde tracers or genetic axonal labelling (intersectional).

In line with the reviewer's question we have now addressed this issue directly by retrograde labeling experiments. Fluorescent beads were thus injected into precise locations in the adult Pitx3-eGFP/wt brain (Supplementary Figure 8). Beads were injected in known targets of SNc and VTA neuronal projections - dorsolateral striatum, nucleus accumbens, and prefrontal cortex. As expected, projections from these areas were detected in regions of the ventral midbrain expressing high levels of TH in SNc and VTA, corresponding to the mapped location of Dat^{high} sublineages. In addition, few labelled cells were in the more medial cells expressing low levels of TH, when beads had been injected in prefrontal cortex and nucleus accumbens. In addition, in the recently published study from the Awatramani group (Poulin et al., Nat Neurosci, 21, 1260) genetic intersectional labeling was used to map the projections of the cells corresponding to the Dat^{high} groups. Our own and previously published tracing is consistent with these results and they are now referred to in the revised manuscript on page 13.

It will be interesting and important to understand more about the projection patterns and function of the Dat^{low} neuron groups defined in this study. As we saw only scattered TH-low cells projecting to VTA and PFC, it is quite possible that their main projection area is somewhere else. Based on an earlier tracing study (Gasbarri et al., Brain Res., 668, 71, 1994) we speculated that TH negative cells in medial VTA, projecting to hippocampus, which were identified in this study, could be cells belonging to our Dat^{low} lineages. However, our own hippocampal tracing experiments showed labelling only in Pitx3-eGFP negative cells in VTA.

Further analysis will essentially require the use of unbiased approaches and preferably virus-based tracing strategies combined with genetic tools to achieve conclusive and detailed understanding of projection patterns. Thus, since the main targets of the Dat^{low} cell groups are unknown, it will be essential to develop a Pitx3-Cre mouse line for the genetic labeling of axons originating from these neurons. We are currently beginning to address these questions, but it will take at least two years to complete these studies and it is impossible to provide conclusive data within the time-frame of the revision.

Minor points:

In the result section, the authors indicate that AT-Dat-high cells were located mainly in the dorsal tier of SNc but based on Fig 4, supplemental fig 4b and report from Poulin et al 2014, AT-Dat high population does not seem to be located in the dorsal SNc but rather in the ventral SNc.

There is some confusion in the literature regarding the question of how to classify this region. For our classification we used the study from George Paxinos (Fu et al., 2011, Brain Structure and Function) who refer to the densely packed DA neurons in the SNc as being localized in the dorsal tier. However, several other investigators are instead considering the region as the ventral tier. To avoid confusion we have revised the manuscript and now refer to the region as SNc.

What is the difference between fig 2b last panel and fig3a?

They are identical. It is a schematic indication of the defined cell groups and the schematic is included again in Fig. 3a for clarity. To avoid misunderstanding we now point out that the two figure items are identical in the legend to Figure 3.

Fig 6c, it is indicated TH low and TH high but they do not show TH staining.

We agree that this looks a bit confusing. The regions correspond to those indicated in Fig. 6a (a' and a" w7) but are not the identical sections. To clarify we now indicate the regions as "medial" and "lateral" in the figure and point out in the legend that they correspond to the regions in Fig. 6a at w7 (and the corresponding regions at w9) expressing high and low levels of TH.

In the discussion they show results about transcription factors but it should be moved in the result section.

The section relating to the developmental transcription factors and Lmx1a-CreERT2 lineage tracing has now been moved to the Results section. It is also briefly mentioned in the Discussion.

Reviewer #2 (Remarks to the Author):

The manuscript by Tiklovà et al. reports an excellent study aimed at identifying and characterizing Pitx3-expressing neuronal subgroups in ventral midbrain.

In the first part of this study, the authors applied scRNAseq to 1106 Pitx3eGFP cells isolated from embryonic and post-natal brains. The authors first analyzed a reliable set of variably expressed genes to cluster these cells on the basis of developmental stage and then analyzed the expression of genes with distinct temporal expression profiles. Both analyses successfully confirm temporal segregation and stage-specific expression. Then the authors analyzed the scRNAseq data to generate a cellular network map where they integrated developmental stages with transcriptional profiles. They generated a map initially distinguishing sorted cells in those with high or low levels of Dat and then, by integrating the expression of additional genes, the authors identified 7 neuronal subgroups.

Immunohistochemistry analysis in adult brain with markers expressed in the identified subtypes allowed their physical map within the Pitx3-expressing domain. Importantly, based on their topology the authors correlated these neuronal subtypes to specific functions. Then, the authors showed that the subtypes they have identified in the adult brain emerged during embryonic development sometimes at different stages. In this context they also reported the subtype-specific expression of several transcription factors which might induce cell-type specific molecular identity during embryonic development. The next step was to investigate the phenotype of Pitx3KO mice showing the precise cell abnormality of this mutant. In the last section of this study the authors reported that neuronal sub-groups with a similar identity could be identified also in the human brain.

In my opinion, this manuscript reports an excellent study. This evaluation is based on the logic approach, methodology employed and general relevance of findings. The manuscript includes an exhaustive amount of data aimed together at the identification of Pitx3-expressing (prevalently dopaminergic) neuronal-subtypes, at their ontogenesis and, further, at their correlation with specific functions. The establishment of this detailed neuronal map should open new avenues which should provide relevant advance to understand development, identity and functioning of specific dopaminergic cell sub-types. This may be important for human diseases related to altered functioning of mDA neurons. In sum I fully support the publication of this manuscript in Nature Communications.

We thank the reviewer for valuable comments.

Reviewer #3 (Remarks to the Author):

In their manuscript titled “Single-cell RNA Sequencing Reveals Midbrain Dopamine Neuron Diversity Emerging During Mouse Brain Development”, Tiklova et al. use single cell RNA-seq to identify midbrain dopaminergic (mDA) cell types during mouse development. This study provides genome-wide gene expression information for an important class of neurons in the mammalian brain along with a public web viewer to mine that data. This study builds on a single cell RNA-seq survey of this same brain structure described by La Manno et al. 2016 in Cell by profiling more mDA cells and including an adult mouse (P90). The authors report a set of cell type markers that persist through development, which should aid targeting these cells. However, the authors missed an opportunity to better relate their results to this previously published work as commented on below. Also, more work needs to be done to show that these same types have conserved markers in human and can be found at equivalent developmental stages.

Major comments:

- You argue that there is a conserved molecular program between mouse and human mDA neuron development but you are comparing cells from earlier in brain development in human. Week 7 is approximately E12.5 in mouse and Week 9 is E14 (see La Manno et al. 2016 Cell, Figure 2B). This is particularly problematic for the expression of Aldh1a1 which turns on in human at week 9 (E14) while in mouse is still off at this stage. You could profile human fetal tissue at matched developmental stages to mouse or you could test for spatial segregation of additional subgroup markers that are persistent throughout mouse development (and hence may be turned on as early as 7-9 weeks in human).

We thank the reviewer for valuable comments.

We agree with the Reviewer that a more detailed analysis of the human mDA neuron development would be interesting. However, as mentioned in the manuscript it was not possible to perform a more comprehensive analysis of human expression patterns since the available tissue was prepared in such a way that in situ hybridization experiments were impossible. Furthermore, several antibodies that we successfully used to identify subgroups in mouse, such as those recognizing Nfib or Slc6a3, did not give any specific signal in the human embryo. However, we showed a similar segregation of Pitx3-positive TH-high and TH-low cells as were found in mouse (Fig. 6a). In addition, by showing expression of ALDH1A1 and EBF3 expression at week 7 and week 9 in human embryonic ventral midbrain it was possible to demonstrate similar temporal gene expression dynamics in both species. We are a bit uncertain about the comment that Aldh1a1 would not be expressed in mouse at E14. Aldh1a1 is expressed in a few lateral cells already at E13.5 in the mouse (see e.g. Fig. 5a). Thus, it is clear that compared to ALDH1A1, EBF3 is expressed more broadly and earlier, but is markedly decreased later in lateral neurons whereas the expression of this marker remains high in the medial region. In turn, ALDH1A1 is induced slightly later and only in the lateral PITX3-expressing cells. This resembles the temporally regulated expression seen in mouse.

- In the discussion, you state that it was difficult to compare your clusters to two existing datasets from Poulin et al. and La Manno et al. This is a missed opportunity that would further validate the diversity of cell types reported in those studies and potentially reveal more diversity. There are a few techniques available to compare samples from disparate datasets – for example, CCA in the Seurat package from the Satija lab (https://satijalab.org/seurat/immune_alignment.html). You could also take a simpler strategy of correlating the expression of cluster marker genes as shown in La Manno et al. (Figure 2A).

As indicated above in our response to Reviewer 1 we now discuss how our groups compare to those described by Poulin et al. (page 19).

In line with the reviewer’s suggestion we also compared our clusters to those reported by La Manno et al. by correlating similarity between clusters (see text on page 19 and new Supp Fig 13). The results show that the five groups of La Manno et al. are all quite similar to two of the Dat^{high} groups in our study (AT-Dat^{high} and T-Dat^{high}). Our other groups are not highly similar to those of La Manno et al. It is not surprising that the Dat^{low} groups do not correspond to those analyzed by La Manno considering that they, similar to Poulin et al., focused on cells expressing high levels of Th and Dat. As mentioned in the response to Reviewer 1, our approach to use Pitx3-

eGFP line captures also cells which lack high levels of Dat and which remain unlabeled by Dat-Cre (referred to as Dat^{low} cells in our manuscript). However, it is a bit surprising that their VTA-3 group, suggested to correspond to the Poulin Vip-expressing DA-2D group, is not similar to the VT- Dat^{high} group in our study. La Manno et al. did not provide information on how the five groups were classified and clustering data is not shown in the paper. It is therefore unclear how distinct these different groups are from each other. It is also difficult to understand their relationship to the data in the Poulin paper. For example, we analyzed the expression of Vip in the cells assigned to the DA-VTA3 group and were surprised to note that only 2 of 26 cells in this cluster expressed any Vip. It is thus somewhat confusing why the DA-VTA3 group is considered to correspond to the Vip-expressing group in Poulin et al. and why it is considered positive for Vip (as indicated in their Figure 6a). We report the comparison between the groups in the revised manuscript as supplementary data (Supplementary Figure 13). However, a comprehensive and conclusive discussion on the comparison between our groups and the groups described by La Manno et al. would be difficult given that clustering data is not reported in their paper and no lists indicating enriched genes for the different groups are available.

Minor comments:

- Mention that you are studying mouse in the abstract

This is stated in the abstract.

- You should report RNA-seq QC metrics: distributions of mapped reads and gene detection

This has now been added as Supplementary Figure 2.

- Please comment on cell type selection biases that may have been introduced by single cell dissociations. What proportion of Pitx3+ cells do you label in situ using markers for your 7 subgroups?

This question is not possible to answer from the histological analysis used in the manuscript since it was limited by the capacity to only combine 1-3 markers at a time. We have now attempted to address this question by using a novel approach for simultaneous analysis of 49 markers by in situ sequencing, using so called Padlock probes (Ke et al., 2013; Supplementary Fig. 7, page 13 in text). These 49 markers were selected on the basis of their enrichment in the various Dat^{low} and Dat^{high} cell groups. Strikingly, the expression profiles resulted in assignments of cell groups that closely matched those defined by scRNA-seq and the previous histological analysis.

In the Supplemental Figure 7 we also provide the proportion of cells assigned to the different groups. When these numbers are compared to the apparent sizes of the groups seen by scRNA-seq (in tSNE in Fig. 2b), it is evident that these two methods give different types of biases between some of the groups. The two biggest groups that we can clearly detect both with traditional ISH and IHC and with Padlock probes, namely AT- Dat^{high} and T- Dat^{high} , are drastically under-represented in scRNA-seq results, most likely due to technical difficulties in the dissociation of mature dopaminergic neurons. Especially the SNC neurons appear particularly vulnerable to dissociation despite numerous efforts to optimize the protocol. Conversely, this dissociation bias results in over-representation of all Dat^{low} groups in the sc-RNA-seq results - these cells, mostly located in the midline of VTA and clearly also smaller in size than Dat^{high} cells, appear more robust and are likely able to withstand better the dissociation protocol. However, in the tissue these Dat^{low} cells appear to comprise a much smaller proportion of the Pitx3-eGFP neurons, when detected both with traditional methods and with Padlock probes. As in situ seq data set closely matches our dataset with traditional ISH and IHC methods, and taken into consideration the above-mentioned technical issues with the tissue dissociation, we think that the percentages of cells assigned to different groups using Padlock data is a quite realistic estimation (now described on page 13). Taken together, based on these analyses, our best answer now is that we think that the great majority, if not all, of Pitx3-eGFP cells can be assigned to one of the seven sublineages also in situ.

- (Figure 1b) You show limited separation of cells at E13.5 and E15.5, and in the Methods you state that variable genes were only selected from 3 ages (E18.5, P1, and P90) to generate the PCA. You should state this in the results that this could explain why you don't see more differences between these two ages. You could also use a method that does not rely on spike-ins to select variable genes from all ages for PCA.

We selected variable genes based on spike-ins that we unfortunately did not have in all stages. But we have done analyses using several different ways of selecting gene sets, and in all cases the separation of the embryonic stages was less clear than the separation later in development. Hence, we do not believe that the selection of variable genes with a different method will have a big effect on the resulting PCA and tSNE.

- Please explain in more detail how you selected the marker genes to combine clusters from Infomap. Based on the graphs you plot and the Infomap results, it looks like cells from different ages should form their own clusters. Do each of your 7 subgroups contain a similar number of Infomap clusters. If you run Infomap with more than 5 nearest neighbors (which is quite low), do you get fewer clusters that better match the final 7 subgroups?

The marker genes were selected based on top loading genes from the PCA as shown in Supplementary Fig 4c. As would be expected, the distinct ages (embryonic, perinatal and adult) clearly separate into different infomap clusters, as can be seen in Supplementary Figure 4d. These clusters were annotated as belonging to one of the main lineages based on gene expression of the top loading genes. In addition, we manually annotated the Th+/- cells. This is described in the Methods section.

We did perform analysis with different number of neighbors in the graph and found the results to be quite robust. With higher number of neighbors we get the same number of infomap clusters, and most cells are assigned to the identical clustering. But in some cases the exact split of the clusters differs and a larger cluster is split into two with one setting but not the other. This is visualized in the image below, with the clustering results using 3 different values for number of neighbors (k). Cells are ordered by clustering with k=5 from largest to smallest cluster and colored in each row by clustering results with different k's.

Number of Infomap clusters in the subtypes:

AT-Dat^{high}	11
T-Dat^{high}	10
VT-Dat^{high}	2
N-Dat^{low}	9
NT-Dat^{low}	3
G-Dat^{low}	8
GT-Dat^{low}	2

• (160-161) If eGFP mRNA levels did not reflect protein levels, then why did you exclude cells on the basis of low eGFP expression before analysis? You showed that some GFP+ cells were PITX3- in situ but you have not shown that these cells correspond to the cells that you filtered. You could find a marker gene of the excluded cells and co-label with this marker in situ.

After clustering we found one cluster with low eGFP expression with high Olig1 expression (Supplementary Figure 1e). We considered those cells as oligodendrocytes and excluded them from analysis (now explained in the Methods). There are still many cells included expressing low levels of eGFP transcript, but they were kept for analysis since they did not form a separate cluster from the remaining cells.

• (235-236) You point out different distributions of G-Dat_{low} and GT-Dat_{low} subclusters in the results, but in Figure 4g you group them together. Can you show in the figure the different distributions or else clarify in the legend text that this is not possible?

We have now added GT-Dat^{low} sublineage in Figure 4g.

• (285-286) In SI Figure 8j you show a massive reduction in AT-Dat_{high} cells by in situ staining but in SI Figure

8d, this reduction looks more modest. How do you explain this difference between RNA-seq profiling and in situ labeling?

With both RNA-seq profiling (now in Supplementary Fig. 11d) and in situ labeling (Supplementary Fig 11e) we can detect some remaining AT-Dat^{high} cells. We did the correction in the cartoon (Supplementary Figure 11j) to visualize those data properly. It should be noted that the cells we show in Supplementary Fig. 11e are the only cells detected in the adult mutant midbrain. Our intention was to show that there still are some Aldh1a1-expressing cells, but indeed very few.

REVIEWERS' COMMENTS:

Reviewer #1 (Remarks to the Author):

The authors have significantly improved their manuscript since the initial submission. The manuscript is thus now suitable for publication in Nature Communications.

Reviewer #2 (Remarks to the Author):

My comments related to the first submission of the manuscript by Tiklovà et al. were already very positive. The additional data provided in the revised version further support the relevance of this study to the understanding of the dopaminergic system.

Reviewer #3 (Remarks to the Author):

The authors have addressed my concerns, and the addition of multiplex FISH to validate cell type distributions strengthens their conclusions. This manuscript provides a helpful contribution to our understanding of neuronal diversity in mouse midbrain.

Minor comments:

- SI Figure 2 – Please label axes with units. For example, why are gene detection values < 1 ?
- SI Figure 8 – It would help to add a table summarizing the projection targets of the 7 Dat-high and Dat-low cell types described in other figures.

Responses to reviewers' comments:

Reviewer #3:

Minor comments:

- SI Figure 2 – Please label axes with units. For example, why are gene detection values < 1?

The figure has now been revised with new x-axis labels.

- SI Figure 8 – It would help to add a table summarizing the projection targets of the 7 Dat-high and Dat-low cell types described in other figures.

We have now added a table to SI Figure 8 according to the reviewer's suggestion.